# Cross-Lagged Associations between Depressive Symptoms and Response Style in Adolescents

**DOI:** 10.3390/ijerph17041380

**Published:** 2020-02-21

**Authors:** Kim M. van Ettekoven, Sanne P. A. Rasing, Ad A. Vermulst, Rutger C. M. E. Engels, Karlijn C. M. Kindt, Daan H. M. Creemers

**Affiliations:** 1Child and Adolescent Psychiatry, GGZ Oost Brabant, P.O. Box 3, 5427 ZG Boekel, The Netherland; 2Erasmus School of Social and Behavioural Sciences, Rotterdam, Erasmus University Rotterdam, P.O. Box 1738, 3000 DR Rotterdam, The Netherlands; 3Child and Adolescent Studies, Utrecht University, P.O. Box 80125, 3508 TC Utrecht, The Netherlands; 4Academic Anxiety Centre, Altrecht, P.O. Box 85314, 3508 AH Utrecht, The Netherlands; 5Behavioral Science Institute, Radboud University, P.O. Box 9104, 6500 HE Nijmegen, The Netherlands

**Keywords:** depression, adolescents, response style, emotion regulation, cross-lagged model

## Abstract

Depressive disorders are highly prevalent during adolescence and they are a major concern for individuals and society. The Response Style Theory and the Scar Theory both suggest a relationship between response styles and depressive symptoms, but the theories differ in the order of the development of depressive symptoms. Longitudinal reciprocal prospective relationships between depressive symptoms and response styles were examined in a community sample of 1343 adolescents. Additionally, response style was constructed with the traditional approach, which involves examining three response styles separately without considering the possible relations between them, and with the ratio approach, which accounts for all three response styles simultaneously. No reciprocal relationships between depressive symptoms and response style were found over time. Only longitudinal relationships between response style and depressive symptoms were significant. This study found that only depressive symptoms predicted response style, whereas the response style did not emerge as an important underlying mechanism responsible for developing and maintaining depressive symptoms in adolescents. These findings imply that prevention and intervention programs for adolescents with low depressive symptoms should not focus on adaptive and maladaptive response style strategies to decrease depressive symptoms, but should focus more on behavioral interventions.

## 1. Introduction

Depression is a serious mental health problem with severe consequences for individuals and society. Almost 30% of the population experiences one or more depressive episodes in their life that have a high cost to society [1]. Regarding adolescents, national and international studies report prevalence rates that vary from 2% to 5.6% [2,3]. Moreover, rates of depression rise dramatically during adolescence. The lifetime prevalence of depression is estimated at 20 to 25% at the age of 19 in the general population [4,5]. Longitudinal research showed a large increase in depressive symptoms in adolescents between ages 13 and 18 [6]. Often, depressive symptoms start to develop in adolescence, with the onset of a depressive disorder in adolescence and young adulthood [7,8]. 

Early adolescence seems to be a critical phase for the development of depressive symptoms. During adolescence, youngsters have to cope with several social, emotional, physical, and emotional challenges [9,10]. Moreover, adolescents experience more stressful life events in comparison with children, but the emotion regulation strategies and the cognitive capacities to cope with these stressors are not fully developed [10]. Additionally, earlier research shows that physical changes in puberty tend adolescents to feel more discomfort about their bodies and become they more insecure and have lower self-esteem [11,12,13]. The vulnerability model postulates that this lower self-esteem could be an important risk factor for the onset of depressive symptoms [14,15]. 

Depression involves disturbances of emotion and experiencing negative mood and a loss of interest or pleasure, which interrupts multiple areas of life, such as daily work, performance at school, and social activities [16,17,18]. The consequences of a depressive disorder, but also elevated depressive symptoms, can be detrimental, leading to poor academic functioning, school drop-out [17], increased risk of suicide [19], mental health problems later in life [20], and substance abuse [21]. Therefore, it is important to understand the underlying mechanisms that are responsible for depressive symptoms to understand the development of depression or depressive symptoms and to intervene in an early stage and to provide the best odds for successful prevention. 

Knowledge of the etiology of depression has increased, although it remains difficult to predict who develops depressive symptoms. One of the key factors involved in the development of depressive symptoms is the way that people respond to their emotions [22,23,24]. In line with this, the Response Style Theory suggests that the way individuals respond to depressive mood influences the duration and severity of their depressive symptoms. This theory proposes three different response styles: rumination, distraction, and problem-solving. First, rumination refers to repetitively focusing on negative feelings and their implications. Second, distraction involves engaging in positive and pleasurable activities to retreat from negative emotions. Lastly, problem-solving involves the tendency to modify someone’s negative mood by actively changing the unfavorable situation [25,26]. Nolen-Hoeksema (1991) described that rumination increases depressive symptoms, whereas distraction and problem-solving decrease depressive symptoms. 

Previous research on the Response Style Theory has mainly focused on adults [25,26,27,28], and only a few studies have focused on adolescent populations [29,30]. A meta-analytic review of Aldao, Nolen-Hoeksema, and Schweizer (2010) showed that, in adolescents, rumination predicts increases in self-reported depression symptoms and the onset of major depression one to four years later [31,32,33]. Previous studies on distraction and problem-solving in adolescents showed ambiguous results in their relationship with depressive symptoms, with some studies reporting that higher levels of distraction were associated with lower levels of depression. One study only found this association for seventh graders [34,35], while other studies found no association or only an association for third graders [30,35]. Regarding problem-solving, studies reported that higher levels of problem-solving were associated with lower levels of depressive symptoms [30,35,36]. Importantly, these studies found no evidence for the predictive value of distraction and problem-solving in relation to the onset of depressive symptoms in children and young adolescents [30,35]. It is important to further examine whether response styles are adaptive or maladaptive in response to emotions and therefore the development of depression because of these mixed findings in previous research and the lack of studies targeting adolescents. 

In contrast to the Response Style Theory, the Scar Theory postulates that depressive symptoms arise first and lead to the use of more maladaptive response styles [37]. It is suggested that individuals with a major depressive disorder acquire several negative psychological characteristics that leave a ‘scar’, defined as a maladaptive response, after recovering from the depressive disorder. As a result, these individuals remain vulnerable to future depressive episodes. Nolen-Hoeksema, Stice, Wade, and Bohon (2007) examined the reciprocal relationship between depressive symptoms and a ruminative response style in 11 to 15-year-old female adolescents. Their study showed that higher depressive symptoms lead to more rumination, which makes these young adolescents vulnerable to experiencing higher levels of depressive symptoms. These young adolescents develop a reciprocal strategy, in which depressive symptoms lead to the development of a ruminative response style, and vice versa [33]. However, in another study, Beevers, Nolen-Hoeksema, Rohde, and Stice (2007) found no evidence for a ruminative response style after recovering from a depressive disorder in 11 to 15-year-old girls, because rumination was elevated before, during, and after the depressive episode. Their study suggested the opposite direction, indicating that more rumination is a chronic risk factor for developing a major depressive disorder [38]. In conclusion, several studies have shown mixed results regarding the Scar Theory. Therefore, it is important to further study the associations between depressive symptoms and the development of a ruminative response style in adolescents. 

The traditional approach to studying the relation between depressive symptoms and response styles is to separately examine the three different response styles, without considering the possible relations between them [29]. For instance, someone can report a high level of rumination that can be compensated by high levels of problem-solving and distraction. The traditional approach could lead to contradictory predictions for the same individual [29]. In contrast, the ratio approach considers all three response styles. The ratio scores of response styles are determined by dividing an individual’s mean rumination score by the sum of the mean problem-solving and mean distraction scores. Consequently, high ratio scores on rumination demonstrate a greater use of rumination to cope with emotion in proportion to problem-solving and distraction. Low ratio scores on rumination indicate a lower use of rumination in proportion to problem-solving and distraction. In this study, we compared both approaches (i.e., the traditional approach and the ratio approach) to compare our findings with previous studies that used the wither ratio- or traditional approach. 

This study aimed to investigate the reciprocal relationship between depressive symptoms and response styles in adolescents over time. In this study, a long-term longitudinal design was used in contrast to most previous studies, which used a short-term longitudinal design (i.e., shorter than 12 months) [28,29,30,34,35,36]. Not many other studies used a long-term longitudinal design (i.e., more than 12 months) [27,31,33]. To our knowledge, just one study investigated the long-term reciprocal relationship between depressive symptoms and response style in adolescents [33]. The longitudinal and reciprocal design can help us to understand the long-term prospective relationships between depressive symptoms and response style over time and, therefore, the development of depressive symptoms in adolescents. The expectation is to find evidence for the Scar Theory, indicating that depressive symptoms will lead to a more ruminative response style, and find evidence for the Response Style Theory, i.e., rumination will lead to more depressive symptoms and distraction and problem-solving will lead to a decrease in depressive symptoms. The expectation is to find an association in both directions, because young adolescents could develop a reciprocal strategy, in which depressive symptoms lead to the development of a ruminative response style and vice versa [33]. This study used the traditional approach as well as the ratio approach to analyze the data to more thoroughly examine the reciprocal relationship between depressive symptoms and response style. 

## 2. Materials and Methods 

The data were derived from a large longitudinal research project aimed at investigating the effects of a selective school-based depression prevention program [39]. A total of 1,343 adolescents participated in a randomized controlled study. The local Ethical Committee of the Radboud University approved the study (ECG13042011).

### 2.1. Participants

The participants that were assigned to both the intervention and control conditions were included in the analyses. Overall, 667 adolescents (49.7%) were assigned to the intervention condition and they received a 16-week depression prevention program ‘Op Volle Kracht’ (OVK), in English ‘On Full Power.’ In OVK, the adolescents were taught cognitive-behavioral techniques to counter negative thinking and improve social and coping skills [40]. A total of 676 adolescents (50.3%) were assigned to the control condition (care as usual). Adolescents in the control condition received the regular school curriculum. All of the participants were in the first or second grade of secondary education from vocational training up to pre-university level. The mean age was 13.4 years (SD = 0.77) and 52.3% were girls. We decided to include participants from the intervention and the control condition in the analyses, because the intervention condition had no significant effect on depressive symptoms in comparison with the control condition [39]. The possible differences between treatment and control group (condition) with respect to baseline covariates (for a description of the covariates, see Section 2.3.3.) were tested with logistic regression analysis. ANOVA was used for the covariate age. Gender, age, ethnic background, parental psychopathology, and school level showed no significant difference between both condition groups. Parental psychopathology showed a significant difference (*p* < 0.05) with a greater number of adolescents who had parents with psychopathology in the control group. For more details see Table 1 in Kind et al., (2014) [39]. Another aspect of combining the intervention and control group to one sample is the degree to which the means, variances, and covariances of both groups can be assumed to be equal. In the statistical analysis in Section 2.4 we tested possible differences between both groups. The results show that equality of means and (co)variances is plausible.

### 2.2. Procedure

Only schools in low-income areas were selected to participate in the study. These schools were defined as schools with at least 30% of their students living in low-income areas in the Netherlands. A total of eleven schools with a total of 57 classes were willing to participate in the study. Classes were randomly assigned to the intervention or control condition. Contamination effects between the intervention and the control group could occur, but the original study tried to minimalize these effects by delivering the OVK program to entire classes instead of groups of randomly assigned individuals or parts of classes. Allocation was performed before the baseline assessment was administered. The parents of the participating students gave passive consent. The exclusion criterion was when parents did not give permission to participate in the study. Data from these students were not collected, but attendance to the program was obligatory because the program was included in the regular school curriculum. When adolescents experience severe depressive symptoms, they were free to visit a psychologist or other care as usual. The extent to which participants received other mental health care during the course of the study was administered. Data were collected by self-reported online questionnaires filled out by the adolescents during class at four time points six months apart, baseline (T1); six months later, after receiving OVK or regular school curriculum (T2); six months follow-up (T3); and, at 12 months follow-up (T4).

### 2.3. Measures

#### 2.3.1. Depressive Symptoms

Depressive symptoms were measured with the Children’s Depression Inventory (CDI) [41]. The CDI is a self-report questionnaire consisting of 27 items. Each item contains three statements, and the participants must choose which statement applied to them the best in the last two weeks. Statements were rated on a three-point scale (e.g., ‘I am sometimes sad’ = 0, ‘I am often sad’ = 1, and ‘I am always sad’ = 2). Item 8 measures the existence of suicidal thoughts, which was eliminated in the study [39]. The total score on depressive symptoms was based on the sum of all items, ranging from 0 to 52 in this study. The higher the total score, the more depressive symptoms will be experienced by the adolescent. The internal consistency and validity of the CDI were good [42]. Cronbach’s alpha coefficients in the current study were 0.85 at T1, 0.88 at T2, 0.90 at T3, and 0.90 at T4.

#### 2.3.2. Response Style.

The Response Style was assessed with the Children’s Response Styles Questionnaire (CRSQ) [35]. The questionnaire contained 25 items measuring individuals’ responses to depressive symptoms. The participants were asked to specify how often they respond in certain ways on a four-point scale (e.g., ‘Almost never’ = 1, ‘sometimes’ = 2, ‘often’ = 3, and ‘almost always’ = 4). The questionnaire contains three subscales: ruminative responses, problem-solving responses, and distracting responses. The scale assessing ruminative responses consists of 13 items, the scale assessing problem-solving responses consists of five items, and the scale assessing distraction responses consists of seven items. Higher sum scores represent a greater tendency to engage in those particular responses to depressive symptoms. Previous studies showed moderate to high levels of internal consistency for all three subscales [30,35]. Cronbach’s alpha coefficients in this study were 0.91 at T1, 0.93 at T2, 0.93 at T3, and 0.94 at T4 for rumination; 0.67 at T1, 0.72 at T2, 0.73 at T3, and 0.75 at T4 for distraction; and, 0.78 at T1, 0.80 at T2, 0.83 at T3, and 0.84 at T4 for problem-solving.

#### 2.3.3. Control Variables

The background variables of the adolescent were derived by single questions about gender, age, and school level. Ethnic background was measured by asking the adolescent in which country they and their parents were born. It was labeled as ethnic background when one of the parents or the adolescent was not born in the Netherlands. Parental psychopathology was measured by the question if one of the adolescents’ parents was treated by a psychiatrist. If a psychiatrist treated one of the parents, the adolescent was analyzed as youth with parent with psychopathology. Condition (yes/no intervention group) was also included as control variable because intervention group and control group were combined in this study.

### 2.4. Statistical Analyses

Regarding statistical analysis, descriptive statistics (means, SDs, correlations) of the research variables, including t-test differences between boys and girls, were conducted first.

Second, we applied structural equation modeling (SEM) while using Mplus 7.2 to examine longitudinal relationships between depressive symptoms and response style [43]. Due to possible non-independence of the data due to nesting of students within (57) classes, we applied the COMPLEX procedure in Mplus to get unbiased estimates of the standard errors of the estimated parameters with the robust maximum likelihood estimator (MLR). To deal with missing data, Full Information Likelihood (FIML) estimation was used with (df) and *p*-value, the comparative fit index (CFI) [44], and root mean square error of approximation (RMSEA) [45] as the fit measures of the SEM models. For CFI, a value > 0.95 is considered good and >0.90 is considered acceptable, while for RMSE, a value < 0.05 means a good fit and a value < 0.08 is an acceptable fit [46].

Coss-lagged modeling was used because we are interested in testing the relationship between depressive symptoms and response style over time. We examined three different relations between latent variables: autoregressive paths between the four adjacent time points for depressive symptoms and the three different response styles separately (i.e., rumination, distraction, and problem-solving), cross-sectional relations between depressive symptoms and response style at each time point, and cross-lagged paths to investigate the relationships between depressive symptoms and response style over time. The cross-lagged models were tested with gender, school level, condition, age, parental psychopathology, and immigration status as control variables. All latent variables of the models were regressed on the control variables. Cross-sectional relations are correlations between the disturbance terms of the latent variables, and they can be interpreted as partial correlations between the latent variables that are corrected for control variables and latent variables of foregoing time points.

Depressive symptoms and the response styles were all treated as latent variables in the model. Using items as indicators of these latent variables can decrease the power of the model (too many parameters to be estimated) [47] and increased estimation problems [48]. For this reason, we created parcels as indicators of the latent variables. The item-indicators of each latent variable were subdivided into subsets of items, according to the “item-to-construct-balance” method [49], and the parcels were computed as the sum of the item-scores of each subset of items. For depressive symptoms, four parcels were created, while three parcels were created for the rumination, distraction, and problem-solving subscales.

Before testing the cross-lagged models, we tested measurement invariance over time for each of the four latent variables to guarantee that the meaning of the latent variables is equivalent over time. First, we tested a longitudinal baseline factor model with factor loadings of the parcels being freely estimated over time. Configural invariance is supported if the fit of the longitudinal baseline model is satisfactory. The second step involved testing metric (weak) invariance by constraining the corresponding factor loadings to be equal over time. Constraining corresponding factor loadings means that the factor loading of parcel 1 at T1 is equal to the factor loading of parcel 1 at T2, T3, and T4. The same applied to the factor loadings of parcel 2, parcel 3, et cetera. The third step involved testing scalar (strong) invariance by constraining the corresponding intercepts to be equal over time. Metric invariance is supported if the decrease of CFI (∆CFI) is <0.01 and increase in RMSEA (∆RMSEA) is < 0.015 in comparison to the configural model [50,51]. In the same way, scalar invariance is supported if, in comparison to the metric model, ∆CFI and ∆RMSEA do not exceed these critical values. Table 1 shows that each of the four latent variables met the criteria for configural, metric, and strong invariances, because ∆CFI and ∆RMSEA are below the critical values for all of the variables. In conclusion, the meaning of the latent variables was equivalent over time. The factor loadings ranged from 0.56 to 0.94 (Table 1), which indicated substantial loadings of the parcels as indicators of all latent variables of the cross-lagged models.

We examined the equality of means and (co)variances of treatment group and control group because the treatment and control group were combined into one sample. We tested whether the means, variances, and covariances of all 52 input variables (16 parcels for depressive symptoms, 12 parcels for rumination, 12 parcels for distraction, and 12 parcels for problem-solving) of the treatment group differed from the control group. The unconstrained model (all parameters estimated free in both groups) was compared with the constrained model (all means, variances, and covariances constrained to be equal in both groups). The chi-square difference test showed a significant difference between both of the groups: ∆χ^2^ (1397) = 2157.30, *p* = 0.000, which indicated a significant difference. However, for large samples, the chi-square (difference) test is always significant and less suited for testing the differences between groups. If the decrease of CFI (∆CFI) is < 0.01 and the increase in RMSEA (∆RMSEA) is < 0.015 in comparison to the unconstrained model, a difference between unconstrained and constrained model is not supported [50,51]. CFI and RMSEA were 0.999 and 0.028 for the unconstrained model, and 0.975 and 0.029 for the constrained model. ∆CFI = 0.024 and support a possible difference between treatment and control group, while ∆RMSEA = 0.001 indicates no difference between both groups. Equality of means and (co)variances of both groups is plausible and both groups can be combined into one sample because the model fit of the constrained model is high (CFI = 0.975 and RMSEA = 0.029).

### 2.5. Drop Out and Missing Vvalues

The study started with 61 classes from 12 schools with N = 1440 adolescents that were randomly allocated to an intervention group (31 classes, N = 741) and control group (30 classes, N = 699). The reasons for drop out of classes were school reasons and teacher’s reasons after allocation to intervention or control group (four classes, N = 91, 6.3%). A second reason for drop out is that adolescents went to another school after T2 (N = 165, 11.5%). Other (incidental) missings are related to the absence of respondents during completion of questionnaires at school. The participation rates were 93.7% (T1), 85.8% (T2), 72.5% (T3), and 74.5% (T4). A full description about drop out and missing values can be found in Kindt et al., (2014) [39].

The first cross-lagged model was tested with depressive symptoms and rumination as the response styles. In the second cross-lagged model, we tested depressive symptoms with distraction and in the third cross-lagged model, and we assessed depressive symptoms with problem-solving. In all models, depressive symptoms and the three response style subscales were treated as latent variables while using parcels as indicators of the latent variables.

We tested depressive symptoms with rumination, distraction, and problem-solving response styles measured on a ratio scale in the fourth, fifth, and sixth cross-lagged models. In this case, the ratio scores for the three response styles were manifest variables. The ratio scores for the response styles were defined as the mean response style 1/(mean response style 2 + mean response style 3).

Cross-lagged paths between depressive symptoms and response style indicate their relationship. Significant cross-paths from depressive symptoms to the response style and the absence of significant cross-paths from response style to depressive symptoms over time can be an indication of a predominant effect of depressive symptoms on response style. Only significant cross-paths from response styles to depressive symptoms can be an indication of a predominant effect of response styles on depressive symptoms. Significant cross-paths in both directions indicate a reciprocal relationship between both of the variables.

## 3. Results

### 3.1. Descriptives

Table 2 presents the means and standard deviations for depressive symptoms from T1 to T4 and the three different response styles (i.e., rumination, distraction, and problem-solving) from T1 to T4. The differences in depressive symptoms and response styles between boys and girls were tested. The effect sizes in terms of Cohen’s d are calculated. Values of 0.2 are interpreted as small effect sizes. Values around 0.5 as medium and values of 0.8 are large effect sizes. Depressive symptoms were significantly higher for girls than for boys only at baseline with low effect sizes. Additionally, girls scored significantly higher on all three response styles across all four waves as compared to boys with low to medium effect sizes.

At the baseline, 79.0% of all adolescents (76.1% girls and 82.2% boys) experienced no problematic depressive symptoms (CDI-score < 13), 13.8% of all adolescents (16.2% girls and 11.1% boys) experienced mild depressive symptoms (cut off score CDI ≥ 13 and CDI < 19), and 7.2% of all adolescents (7.7% girls and 6.6% boys) experienced severe depressive symptoms (cut off score CDI ≥ 19). The association between gender and the three levels of depressive symptoms was significant (χ^2^(2) = 7.91 with *p* < 0.05). Post-hoc testing (z-tests for proportions with Bonferroni correction with critical alpha = 0.05/3 = 0.016) showed a significant underrepresentation of girls in the no problematic group (z = -2.65, *p* = 0.008 and <0.016), a significant overrepresentation of girls (in the mild depressive symptoms group z = 2.62, *p* = 0.009 and <0.016), and no significant difference between boys and girls in the severe depressive symptoms group (z = 0.75, *p* = 0.453 and >0.016).

Subsequently, Table 3 presents the Pearson’s correlations between all of the variables. For the traditional scores (below diagonal), depressive symptoms were positively related to rumination at all four time-points, but their correlations with distraction and problem-solving were very low and mostly nonsignificant. Rumination was significantly positively associated with distraction and problem-solving. These correlations indicate that higher depressive symptoms were associated with higher levels of rumination, higher levels of rumination were related to higher levels of distraction and problem-solving, and higher levels of distraction were linked to higher levels of problem-solving. For the ratio scores (above diagonal), all the correlations of depressive symptoms with rumination were positive, significant, and substantial. The correlations of depressive symptoms with distraction and problem-solving were significantly negative, but rather low. Rumination correlated negatively with distraction and problem-solving and distraction correlated negatively with problem-solving. This means that higher levels of depressive symptoms are indicative of higher levels of rumination, but lower levels of distraction and problem-solving. Higher levels of rumination were related to lower levels of distraction and problem-solving, while higher levels of distraction were associated with lower levels of problem-solving. For traditional scores as well as for ratio-scores, cross-sectional (identical timepoints) correlations were substantially higher. 

Gender, school level, condition, age, parental psychopathology, and ethnic background were used as the control variables in the cross-lagged models. For each cross-lagged model, all variables in a model were regressed on the control variables. At T1, gender showed positive associations with all of the variables used in the cross-lagged models, with the exception of one negative association with ratio distraction. Age was positively related to depressive symptoms and negatively related to distraction and ratio distraction. School level was positively related to ratio rumination and to ratio problem-solving and negatively related to ratio distraction. Parental psychopathology had a negative association with depressive symptoms, rumination, ratio rumination, distraction, and with ratio problem-solving. Finally, ethnic background negatively correlated with ratio problem-solving. See Table A1 and Table A2 for all associations between the control variables and the variables in cross-lagged models at all time points.

### 3.2. Cross-Lagged Models of Depressive Symptoms with Response Styles (Traditional Scores)

Figure 1, Figure 2 and Figure 3 demonstrate the autoregressive paths, cross-sectional paths, and cross-lagged paths between latent variable depressive symptoms and latent variables rumination, distraction, and problem-solving, in which response style was measured with traditional scores.

*Model fit and autoregressive paths.* The cross-lagged models demonstrated a good fit (Figure 1: χ^2^(412) = 679.00, *p* = 0.000, CFI = 0.986 and RMSEA = 0.022; Figure 2: χ^2^(412) = 806.32, *p* = 0.000, CFI = 0.969 and RMSEA = 0.027; Figure 3: χ^2^(412) = 711.50, *p* = 0.000, CFI = 0.979 and RMSEA = 0.023). All autoregressive paths, from T1 to T2, from T2 to T3, and from T3 to T4, were significant. The latent factor depressive symptoms and the latent factors response styles were stable over time.

*Cross-sectional paths.* The cross-sectional partial correlations of depressive symptoms with rumination were significant at all four time points. No significant cross-sectional (partial) correlations between depressive symptoms and distraction at any time point were found, and only one positive significant cross-sectional relation between depressive symptoms and problem-solving was found at T3.

*Cross-lagged paths.* For rumination, we only found a positive significant cross-lagged path from depressive symptoms at T2 to rumination at T3 (β = 0.11, *p* < 0.05). This indicated that depressive symptoms at T2 predicted an increase in rumination six months later. According to Figure 2, a positive significant cross-lagged path was found from distraction at T2 to depressive symptoms at T3 (β = −0.10, *p* < 0.05). No other cross-lagged paths were found in the model between depressive symptoms and distraction. Two negative significant cross-lagged paths were found from problem-solving at T2 to depressive symptoms at T3 (β = −0.08, *p* < 0.05) and from problem-solving at T3 to depressive symptoms at T4 (β = −0.09, *p* < 0.05).

### 3.3. Cross-Lagged Models of Depressive sSymptoms with Response Styles (Ratio Scores)

Figure 4, Figure 5 and Figure 6 demonstrate the autoregressive paths, cross-sectional paths, and cross-lagged paths between the latent variable depressive symptoms and manifest variables rumination, distraction, and problem-solving, in which the response style was measured with ratio scores.

*Model fit and autoregressive paths.* The models demonstrated a good fit (Figure 4: χ^2^(206) = 391.43, *p* = 0.000, CFI = 0.982, RMSEA = 0.026; Figure 5: χ^2^(206) = 416.06, *p* = 0.000, CFI = 0.979, RMSEA = 0.028; Figure 6: χ^2^(206) = 412.66, *p* = 0.000, CFI = 0.980, RMSEA = 0.027). All of the autoregressive paths in the three models were positive and significant, which suggests stability over time.

*Cross-sectional relations.* All of the cross-sectional paths in Figure 4 were significant and positive. Cross-sectional paths in Figure 5 and Figure 6 were all significant and negative.

*Cross-lagged paths.* For rumination, two positive cross-lagged associations from depressive symptoms to rumination were found from T1 to T2 (β = 0.10, *p* < 0.05) and from T3 to T4 (β = 0.17, *p* < 0.01). Another cross-lagged association was found between rumination at T2 and depressive symptoms at T3 (β = 0.08, *p* < 0.05). Regarding distraction, two negative cross-lagged paths were found from depressive symptoms at T1 and T3 to distraction at T2 (β = −0.12, *p* < 0.00) and T4 (β = −0.11, *p* < 0.01), respectively. For problem-solving, negative cross-lagged associations were found at all time points from depressive symptoms to problem-solving six months later (T1: β = −0.11, *p* < 0.01; T2: β = −0.16, *p* < 0.00; T3: β = −0.12, *p* < 0.01). No reciprocal associations were found between depressive symptoms and any response style.

## 4. Discussion

This study aimed to investigate the longitudinal reciprocal relationship between depressive symptoms and response style over time in a non-clinical adolescent sample. Additionally, two different approaches for measuring response style were used in this study (i.e., traditional approach and ratio approach) to examine this relationship. The expectation was to find reciprocal relationships between the depressive symptoms and response style, which was in line with the Scar Theory and the Response Style Theory.

First, we used two different methods to study the relationship between depressive symptoms and rumination. The results that response style predicted depressive symptoms more consistently in the analyses with ratio scores demonstrated in comparison to the analyses that were executed with the traditional approach. Besides that, the cross-lagged models in the traditional approach only showed consistent cross-sectional associations between depressive symptoms and rumination and no consistent relationships of depressive symptoms with distraction and problem-solving. In the analyses that were executed with the ratio scores, all of the cross-sectional paths were strongly correlated. Moreover, we found stronger cross-sectional and cross-lagged relationships between depressive symptoms and response style in the models with the ratio approach in comparison with the traditional approach. In conclusion, by using two different approaches to analyze the relationship between depressive symptoms and response style, different results were found. The ratio scores consider all three response styles to offer a more reliable overview of the use of a particular response style because of the more consistent relationships found between depressive symptoms and response style using the ratio approach and the fact that from a theoretical view; we focus more on the results derived from ratio scores than from analyses with traditional scores to simplify further interpretation.

In this study, we found consistent longitudinal relationships between depressive symptoms and rumination while using the ratio score in contrast to using the traditional scores, where no consistency in the relationships was found. This finding with the ratio scores is in line with the Scar Theory, which states that, after recovering from depression, a ‘scar’ is left that could lead to a greater tendency to engage in a ruminative response style [37]. Additionally, this study (only for ratio scores) also found negative relationships between depression and distraction/problem-solving. This indicates that depressive symptoms predict adaptive response styles. In conclusion, depressive symptoms were predictive of any response style, although not the other way around, in this study.

We expected to find a positive relationship between a ruminative response style and depressive symptoms over time and a negative relationship between adaptive response styles (problem-solving and distraction) and depressive symptoms over time, according to the Response Style Theory [25]. The results of this study, with both ratio and traditional scores, revealed no longitudinal evidence for the predictive value of rumination on depressive symptoms and of distraction and problem-solving on depressive symptoms, thus supporting the Response Style Theory. The results of the analysis with traditional scores showed some, but limited and inconsistent, evidence for the predictive value of only problem-solving on depressive symptoms over time.

In comparison with other studies that found support for the Response Style Theory, this study could not replicate these findings, which was possibly due to the relatively low depressive symptoms (i.e., M = 8.23 at T1, M = 9.08 at T2, and M = 9.23 at T3 and T4) and rumination (i.e., M = 7.18 at T1, M = 7.53 at T2, M = 7.36 at T3, and M = 7.16 at T4) in this sample. It was not clearly defined how severe and how long depressive symptoms must last for them to leave a ‘scar’, resulting in a ruminative response style afterward. Possibly, the relatively low scores on depressive symptoms in this sample did leave a scar, but not severe enough (because of the low depressive symptoms) to predict depressive symptoms. Therefore, it might be possible to find support for this relationship in a clinical sample, because these adolescents experience higher depressive symptoms. Additionally, depressive symptoms are known to increase during adolescence [52]. The lowest rates of depressive symptoms were found during childhood up to early adolescence. From the age of 13, the start of a trend in the increase of depressive symptoms is seen with a strong increase between 15 and 18 years of age [6]. From adulthood (18 years of age), depressive symptoms become relatively stable [53]. This could imply that the relationship between depressive symptoms and response style could differ for adult samples due to the more stable character of depressive symptoms and more increased rates of depressive symptoms from adulthood. Furthermore, other studies finding prospective relationships between rumination and depressive symptoms used shorter follow-up intervals (i.e., three weeks, six weeks, and four months) [30,34,36]. In this study, a six-month time interval was used, which could be too long to find prospective relationships between response style and depressive symptoms. We do not exactly know which time interval is needed to find changes in adolescents’ experienced depressive symptoms following the use of a ruminative response style, and vice versa. From another point of view, the used the six-month time interval with the longest follow up measure at 18 months could be too short to detect an effect when we keep in mind that the strongest increase in depressive symptoms is found between the age of 15 and 18 [6]. The mean age of the adolescents in our sample at the start of the intervention was 13.4 years, which implies that, 18 months later, they were around 15 years, just at the start of the strong increase of depressive symptoms. By using a longer follow up framework that ends more between 15 to 18 years of age, it could lead to other results. Furthermore, scores on depressive symptoms in adolescents highly fluctuate over time; therefore, we might have included too few time points to measure depressive symptoms. Moreover, the longitudinal model in this study consisted of three time intervals to investigate reciprocal relationships, but most of the studies that found prospective relationships between depressive symptoms and response style only used one [30,34] or two time intervals [36] and drew conclusions from fewer follow-up measures to describe the relationship. Our study also found a prospective association between depressive symptoms and response style at one time interval, but not at all time intervals. Using multiple time intervals to describe a relationship might make it more difficult to find consistent prospective relationships over time. However, multiple time intervals should be included to draw substantial conclusions regarding the relationship between depressive symptoms and response style [54]. Therefore, our study with three time intervals provided a more consistent overview of the existing reciprocal relationship between depressive symptoms and response style over time.

Additionally, regarding the results of our study, we possibly have to consider rumination as an important symptom or cognitive trait associated with depressive symptoms, rather than a predictor of depressive symptoms. This is in line with the strong correlations found between depressive symptoms and rumination at all time points in this study and other studies using cross-sectional data [55,56]. These findings suggest that increased rumination led to more depressive symptoms at the same time point, or vice versa. This indicated that, in adolescents, rumination is an important factor that is associated with depressive symptoms and, thus, an important factor to consider in the prevention and treatment of depression.

This study has several noticeable strengths and limitations. One of the strengths is that we used multiple methods to study the relationship between depressive symptoms and response style, a traditional way of measuring response style and ratio scores as response style. Therefore, we could compare the results of the two approaches to measuring response style. Second, the study was conducted with a large sample of adolescents to provide a better understanding of this high-risk population. Third, a longitudinal design was used with three time points and a follow-up period of 12 months to provide more robust information regarding the prospective relationships between depressive symptoms and response style over time.

This study also has some limitations. First, answers could be influenced by social desirability bias by using direct self-report questionnaires [57]. Second, the scores on depressive symptoms were relatively low and only a few adolescents experienced severe depressive symptoms due to the sample used in this study (7.2%; cut off score CDI ≥ 19). This means that our results are based on information regarding the relationship between depressive symptoms and response style in adolescents who experienced few depressive symptoms. The expectation is that severe levels of rumination increase the likelihood of experiencing depressive symptoms, according to the Response Style Theory. Therefore, future research should study the relationship between depressive symptoms and response style in a clinical sample over time. Third, we used time interval periods of six months, but we do not know whether changes in depressive symptoms or changes in rumination occur within six months, given the fact that depressive symptoms in adolescents can fluctuate highly over time. Future studies should include a greater number of shorter time intervals (e.g., Experience Sampling Method) to examine the relationship between depressive symptoms and response style over time [58]. The Experience Sampling Method involves an intensive longitudinal methodology that allows for the participants to report their mood, behavior, feelings, or thoughts in the moment at multiple times a day for serval weeks [58]. Therefore, it allows for us to obtain more detailed insight into the relationship between depressive symptoms and response style over time due to the possible high fluctuations in depressive symptoms in adolescents. Lastly, we must be careful in generalizing these findings to all adolescents, because we used a non-community sample by only including schools with >30% of their students living in low-income areas.

The results of this study offer directions for future research and they have some implications in practice. Future research should focus on other underlying mechanisms (e.g., negative cognitions, other more specific emotion regulation strategies, life events, earlier experienced depressive episodes) rather than only on response style. Moreover, depressive episodes experienced earlier in life and their residual symptoms are an important predictor of experiencing future depressive episodes [59,60]. As described in the Scar Theory, it is relevant to investigate which of these residual symptoms in a clinical sample are responsible for developing future depressive episodes.

Our findings imply that that prevention and intervention programs that target adolescents with low depressive symptoms should not focus on adaptive and maladaptive response style strategies, because our findings indicated no consistent prospective relationship between adaptive or maladaptive response style strategies and the decrease or increase in depressive symptoms over time. Adaptive and maladaptive response styles become more severe after experiencing more severe depressive symptoms, but they do not lead to an increase or decrease in the severity of depressive symptoms. However, the strong correlations between depressive symptoms over time indicate that depressive symptoms in adolescents are quite stable over time. Therefore, it is important to identify adolescents that are at risk for developing depressive symptoms to offer them appropriate treatment to prevent them from experiencing depressive symptoms. The strong cross-sectional relationships between depressive symptoms and rumination suggest that we might be able to identify adolescents at risk based on the extent to which they use a ruminative response style. The treatment of adolescents at risk should incorporate behavioral interventions (e.g., behavioral activation and mood monitoring) rather than cognitive techniques [61]. Eventually, in this study, we only distinguished one maladaptive response style strategy and two adaptive strategies. Future research could include different maladaptive and adaptive response strategies (e.g., adaptive strategies: acceptance, humor enhancement, revaluation, and forgetting; maladaptive strategies: withdrawal, giving up, self-devaluation, and aggressive actions) to learn more about other specific response style strategies that are possibly responsible for changes in depressive symptoms over time.

## 5. Conclusions

In conclusion, the present study found no evidence for the reciprocal relationship between depressive symptoms and response style over time. According to the findings of this study, we could not indicate that response style functions as an underlying mechanism that is responsible for developing and maintaining depressive symptoms in adolescents. This study only found evidence for depressive symptoms as a predictor for response style. More experienced depressive symptoms could leave a ‘scar’ in the form of a maladaptive response, with fewer depressive symptoms leading to more adaptive response styles. These findings imply that prevention and intervention programs for adolescents with low depressive symptoms should focus not only on using maladaptive and adaptive response style strategies to decrease depressive symptoms, but also on incorporating behavioral interventions. Taking the limitations of the used interval period for measuring response style and depressive symptoms and the focus on only three response style strategies into account, future studies should focus on a greater number of shorter time intervals and other types of adaptive and maladaptive response style strategies or other underlying mechanisms that are possibly responsible for developing and maintaining depressive symptoms in adolescents.

## Figures and Tables

**Figure 1 ijerph-17-01380-f001:**
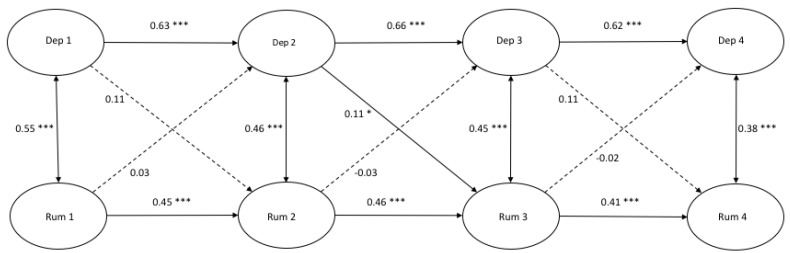
A cross-lagged model of depressive symptoms and rumination. *Note.* * *p* < 0.05 *** *p* < 0.001.

**Figure 2 ijerph-17-01380-f002:**
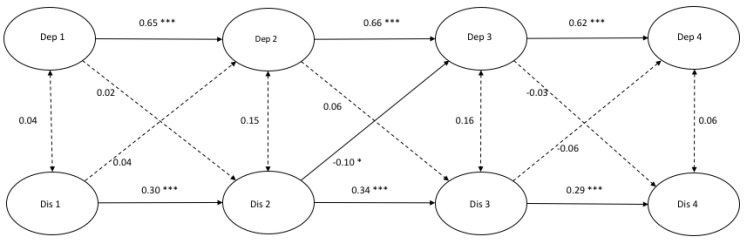
A cross-lagged model of depressive symptoms and distraction. *Note.* * *p* < 0.05 *** *p* < 0.001.

**Figure 3 ijerph-17-01380-f003:**
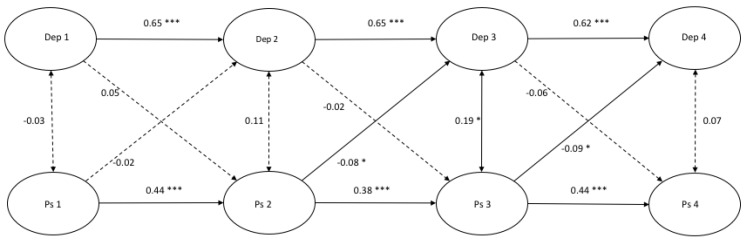
A cross-lagged model of depressive symptoms and problem-solving. *Note.* * *p* < 0.05 *** *p* < 0.001.

**Figure 4 ijerph-17-01380-f004:**
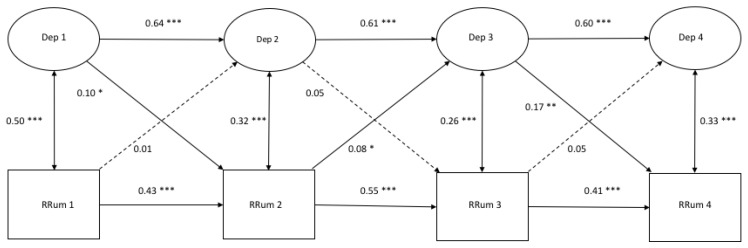
A cross-lagged model of Depressive Symptoms and Rumination measured with ratio scores. *Note.* * *p* < 0.05 ** *p* < 0.01 *** *p* < 0.001.

**Figure 5 ijerph-17-01380-f005:**
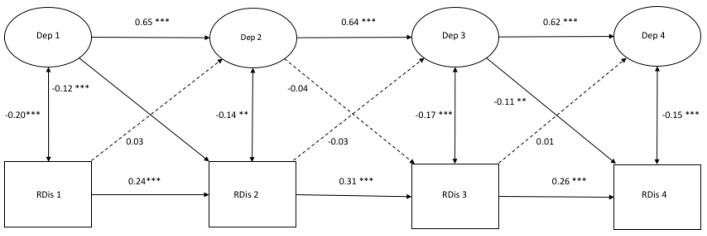
A cross-lagged model of Depressive Symptoms and Distraction measured with ratio scores. *Note.* * *p* < 0.05 ** *p* < 0.01 *** *p* < 0.001.

**Figure 6 ijerph-17-01380-f006:**
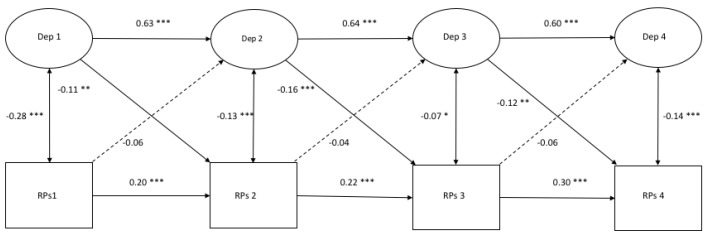
A cross-lagged model of Depressive Symptoms and Problem-Solving measured with ratio scores. *Note.* * *p* < 0.05 ** *p* < 0.01 *** *p* < 0.001.

**Table 1 ijerph-17-01380-t001:** Longitudinal configural, metric, and strong invariance of four latent variables and factor loadings of the parcels.

									Standardized Factor Loadings			
		*χ* ^2^	*df*	*p*	RMSEA	CFI	∆CFI	∆RMSEA	Mean	SD	Min	Max
depressive symptoms	configural invariance	94.86	74	0.052	0.015	0.998			0.82	0.04	0.72	0.87
	metric invariance	111.04	83	0.022	0.016	0.997	0.001	0.001				
	strong invariance	133.84	92	0.003	0.019	0.995	0.003	0.002				
rumination	configural invariance	35.58	30	0.222	0.012	0.999			0.89	0.03	0.85	0.94
	metric invariance	52.49	36	0.037	0.019	0.998	0.007	0.001				
	strong invariance	62.13	42	0.023	0.019	0.997	0.000	0.001				
distraction	configural invariance	40.70	30	0.092	0.016	0.996			0.67	0.08	0.56	0.79
	metric invariance	47.83	36	0.090	0.016	0.995	0.000	0.001				
	strong invariance	63.24	42	0.019	0.020	0.992	0.004	0.003				
problem-solving	configural invariance	37.76	30	0.156	0.014	0.998			0.75	0.08	0.58	0.84
	metric invariance	40.69	36	0.272	0.010	0.999	−0.004	−0.001				
	strong invariance	56.04	42	0.072	0.016	0.996	0.006	0.003				

**Table 2 ijerph-17-01380-t002:** Mean scores, standard deviations, and gender differences.

	Total	Boys	Girls	t-Value	Cohen’s d
T1 Depressive Symptoms	8.23 (6.26)	7.76(6.28)	8.66 (6.22)	2.55 *	0.14
T2 Depressive Symptoms	9.08 (7.42)	8.86 (8.16)	9.27 (6.69)	0.92	0.05
T3 Depressive Symptoms	9.23(7.84)	8.77 (8.29)	9.63 (7.42)	1.69	0.11
T4 Depressive Symptoms	9.23(8.17)	8.94 (8.79)	9.47 (7.61)	1.00	0.06
T1 Rumination	20.18 (7.40)	18.40(6.20)	21.85 (8.02)	8.36 ***	0.48
T2 Rumination	20.53 (7.79)	18.76 (7.00)	22.17 (8.13)	7.40 ***	0.45
T3 Rumination	20.36 (7.79)	18.48 (7.28)	21.89 (7.85)	6.96 ***	0.46
T4 Rumination	20.16 (7.64)	18.16 (6.47)	21.83 (8.13)	7.72 ***	0.50
T1 Distraction	13.24 (3.92)	12.79 (4.01)	13.66 (3.80)	3.90 ***	0.22
T2 Distraction	12.80 (4.00)	12.19 (4.24)	13.35 (3.69)	4.83 ***	0.29
T3 Distraction	12.58 (3.94)	11.81 (4.11)	13.25 (3.65)	5.59 ***	0.37
T4 Distraction	12.39 (3.94)	11.61 (3.98)	13.03 (3.79)	5.73 ***	0.37
T1 Problem-Solving	8.63 (3.20)	7.74 (2.90)	9.46 (3.24)	9.72 ***	0.56
T2 Problem-Solving	8.43 (3.14)	7.87 (3.13)	8.93 (3.06)	5.65 ***	0.34
T3 Problem-Solving	8.39 (3.24)	7.53 (3.11)	9.13 (3.16)	7.67 ***	0.51
T4 Problem-Solving	8.44 (3.29)	7.62 (3.02)	9.13 (3.34)	7.34 ***	0.47

*Note.* * *p* < 0.05 *** *p* < 0.001.

**Table 3 ijerph-17-01380-t003:** Pearson’s correlation coefficients between all the variables included in the models.

	1	2	3	4	5	6	7	8	9	10	11	12	13	14	15	16
1. Dep. symptoms T1		0.58 **	0.53 **	0.41 **	0.48 **	0.30 **	0.29 **	0.22 **	−0.20 **	−0.16 **	−0.17 **	−0.09 **	−0.24 **	−0.15 **	−0.11 **	−0.12 **
2. Dep. symptoms T2	0.58 **		0.57 **	0.45 **	0.32 **	0.40 **	0.26 **	0.26 **	−0.10 **	−0.20 **	−0.11 **	−0.13 **	−0.21 **	−0.20 **	−0.17 **	−0.14 **
3. Dep. symptoms T3	00.53 **	0.57 **		0.58 **	0.27 **	0.30 **	0.37 **	0.31 **	−0.09 *	−0.15 **	−0.20 **	−0.14 **	−0.18 **	−0.16 **	−0.18 **	−0.16 **
4. Dep. symptoms T4	0.41 **	0.44 **	0.57 **		0.21 **	0.24 **	0.25 **	0.42 **	−0.06	−0.13 **	−0.11 **	−0.21 **	−0.15 **	−0.12 **	−0.17 **	−0.21 **
5. Rumination T1	0.50 **	0.34 **	0.30 **	0.18 **		0.48 **	0.39 **	0.29 **	−0.50 **	−0.27 **	−0.26 **	−0.16 **	−0.43 **	−0.20 **	−0.11 **	−0.14 **
6. Rumination T2	0.34 **	0.50 **	0.32 **	0.25 **	0.51 **		0.58 **	0.41 **	−0.25 **	−0.53 **	−0.33 **	−0.24 **	−0.18 **	−0.42 **	−0.21 **	−0.19 **
7. Rumination T3	0.30 **	0.33 **	0.47 **	0.28 **	0.42 **	0.50 **		0.48 **	−0.16 **	−0.31 **	−0.52 **	−0.24 **	−0.21 **	−0.23 **	−0.43 **	−0.23 **
8. Rumination T4	0.23 **	0.27**	0.30 **	0.41 **	0.32 **	0.43 **	0.45 **		−0.11 **	−0.19 **	−0.22 **	−0.49 **	−0.16 **	−0.21 **	−0.24 **	−0.46 **
9. Distraction T1	0.04	0.07 *	0.05	−0.02	0.33 **	0.12 **	0.15 **	0.05		0.27 **	0.25 **	0.18 **	−0.49 **	−0.02	−0.11 **	−0.08 *
10. Distraction T2	0.04	0.11 **	0.01	0.00	0.15 **	0.41 **	0.11 **	0.07 *	0.31 **		0.33 **	0.26 **	−0.01	−0.47 **	−0.01	−0.03
11. Distraction T3	0.03	0.09 *	0.14 **	0.04	0.12 **	0.11 **	0.43 **	0.13 **	0.28 **	0.34 **		0.28 **	0.02	−0.03	−0.47 **	−0.07
12. Distraction T4	0.06	−0.03	0.03	0.04	0.14 **	0.16 **	0.18 **	0.48 **	0.16 **	0.25 **	0.32 **		−0.03	−0.03	−0.06	−0.46 **
13. Problem−solving T1	0.01	−0.01	−0.03	−0.07 *	0.42 **	0.22 **	0.20 **	0.13 **	0.49 **	0.29 **	0.29 **	0.20 **		0.22 **	0.22 **	0.23 **
14. Problem−solving T2	0.05	0.10 **	0.02	0.02	0.23 **	0.50 **	0.17 **	0.16 **	0.24 **	0.56 **	0.25 **	0.24 **	0.37 **		0.25 **	0.25 **
15. Problem−solving T3	0.05	0.05	0.12 **	0.01	0.20 **	0.20 **	0.52 **	0.19 **	0.19 **	0.26 **	0.60 **	0.27 **	0.35 **	0.33 **		0.34 **
16. Problem−solving T4	0.04	0.03	0.01	0.02	0.16 **	0.17 **	0.21 **	0.51 **	0.12 **	0.15 **	0.26 **	0.60 **	0.29 **	0.29 **	0.40 **	

*Note.* ** Correlation is significant at the 0.01 level (2-tailed), * Correlation is significant at the 0.05 level (2-tailed). Correlations are reported for traditional scores of response style below the diagonal and correlations for the ratio scores of response style above the diagonal.

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
