# Peer review of "Cross-Lagged Associations between Depressive Symptoms and Response Style in Adolescents"

_ijerph, 2020, doi:10.3390/ijerph17041380_

Round 1

Reviewer 1 Report

This is an interesting study employing a relatively sophisticated statistical method. 

One point I would suggest is to add at least a brief passage addressing the reasons that the literature has posited for the onset of depressive symptoms in adolescence, and the related references (e.g. Babore, Trumello, Candelori, Paciello et al., 2016). Why is this specific developmental stage particularly at risk for emotional/behavioral vulnerability? 

Another important point, in my opinion, refers to the random allocation of subjects to the two study groups.

If the screening shown clinical problems in this population, ethical constraints would have imposed to propose an intervention to those showing symptoms, not to assign subjects randomly. Maybe I missed some information on this, so please clarify this point. To this end, it would be useful to show the cut-offs of the considered variables.

I suggest including the excluding criteria for the sampling from the larger study.

It is not clear to me which measure was used to asses parental psychopathology. Please clarify.

Due to the nature of the statistics, I suggest adding a specific paragraph to describe the Analysis plan.

Reviewer 2 Report

This article offers a nicely presented analysis of possible theoretical mechanisms behind adolescent depression. A large sample size was used, and longitudinal data collected. The research is theoretically strong and quantitatively sophisticated. I have just a few comments for the authors to consider.

Despite apparently null results of the effect of the intervention, it seems risky to include both the control and intervention groups in this same analysis. I recognize condition was included in models, which is sensible, but it still makes me a bit nervous. Would the authors consider a sensitivity analysis of sorts, perhaps examining at least correlational data in only the control group to be sure it is a reasonable match with the overall sample? I’m surprised the gender difference in depressive symptoms was present only at baseline. Any thoughts as to why it disappeared at later time points? Page numbers and line numbers are mixed up in my version of the manuscript, but the paragraph starting, “Gender, school level, …” examining regressions of control variables is full of lots of results that might be better presented in a table. Given the open-access online format of IJERPH, perhaps a table could be created to show these results? I liked the attention to the developmental aspects of this research – studying adolescents rather than adults – that was presented in the introduction. I was surprised this topic was not addressed again in the discussion. Is it worth mention while interpreting the results? Might results possibly be different among a sample of adults?

Reviewer 3 Report

Review

This study addresses an interesting and important topic by using novel longitudinal data to test two competing theories regarding the development of depressive symptoms in adolescents. Overall, I have found the paper to be well-written and offer useful insights to an important topic. Below I detail some recommended changes that I believe may improve the overall quality of the manuscript.

Major Comments

On line 104, when the authors note that most previous studies were cross sectional can they include the citations here to note which studies were cross-sectional. In addition, the authors note that “most” studies were cross-sectional. However, that language suggests that some studies were not. Can the authors also note here which studies were longitudinal? Doing so can help the readers better evaluate the state of the literature by clearly lining up the cross-sectional versus longitudinal studies.

The participants were recruited from 11 schools and 57 classrooms. While the treatment was administered randomly, given that the participants are coming from a small cluster of schools and classes within those schools, I wonder about any chances of contamination between the control and treatment groups. It seems plausible that some of these children were friends and could have discussed any treatment/experiences they had. Specifically, were there any steps taken to account for or detect contamination across the treatment and control groups?

I am curious about how this two-week time reference for depressive symptoms period jives with T2 data collection window. At T2 data are collected 6-months after baseline and immediately after receiving OVK. That means the depressive symptoms being measured here are asking about symptoms both before and after receiving treatment, if I am understanding this correctly. This may conflate the depressive symptoms immediately prior to the treatment (i.e. two weeks before OVK), with depressive symptoms post treatment (symptoms now). Depending on what reference period participants use, it seems it may be possible that some are reflecting on symptoms one or two weeks earlier, whereas some are reflecting on symptoms now, which could be improved since the treatment was recently administered.

Does the depression scale (25 items) contain all the items from the CRSQ? If so, can the authors state that the entire scale was used.

In regard to the imputation/missing data. I have a few questions: (1) how many cases were missing? (2) what was the major source of missing data? (3) how did attrition from the sample change across T1-T4. (4) Was attrition significantly different across control and treatment groups. (5) How many imputed data sets were created? (6) Are results different if the authors used a listwise deletion sample compared to an imputed sample.

Can the authors include a table (either in the main text or an appendix) that compares the treatment and control group at baseline across co-variates to show whether there were any significant differences in the group following randomization.

The control variables show up for the first time on page 10 of the manuscript. But there is no prior discussion of how they care coded and collected. Can the authors provide some detail of these either in the text or in an appendix about the measurement of these variables?

It seems that age and school level would be highly correlated? Is it necessary to include both these measures?

I appreciate the authors reporting the effect size and significance in the text. However, when beginning to discuss the results it may be helpful to add a sentence or two that interprets the effect size in order to give the readers an idea about what the magnitude of the impact is. This does not need to be done for every result. But could be helpful to be done once at the start of the results section to better help readers evaluate whether the impact is large or small.

In the discussion section, the authors note that a 6-month time frame may have been too long to detect an effect. However, is it also possible that it is too short of a time frame and that there was not enough time for symptoms to change or too small a window to detect substantial variation in symptoms.

Round 2

Reviewer 1 Report

The Authors were responsive to all comments and I think the paper can be published in the present form.

Reviewer 3 Report

Thank you for your revisions.